# Banking Industry Sustainable Growth Rate under Risk: Empirical Study of the Banking Industry in ASEAN Countries

**Isnurhadi** [1], **Sulastri** [1,*], **Yulia Saftiana** [2] **and Ferry Jie** [3]

1   Management Department, Fakultas Ekonomi, Universitas Sriwijaya, Palembang 30128, Sumatera Selatan, Indonesia
2   Accounting Department, Fakultas Ekonomi, Universitas Sriwijaya, Palembang 30128, Sumatera Selatan, Indonesia
3   School of Business and Law, Edith Cowan University, Joondalup 6027, Australia
*   Correspondence: sulastri@unsri.ac.id

**Abstract:** This research examines how the banking industry maintains its sustainable growth rate. The sample consists of 328 commercial banks in the ASEAN area. A fixed effect model is employed to analyze the data. The study reveals several findings: (1) The countries with the most risk in the banking industry are Indonesia, Thailand, Philippines, Malaysia, and Singapore. (2) Operational risk has a negative effect on sustainable growth and a positive effect on actual growth. Asset utilization positively affects sustainable growth and positively affects actual growth. (3) Business risk has a positive effect on sustainable growth but a negative on actual growth. (4) Liquidity risk positively affects both sustainable growth and actual growth. (5) Financial risk has a negative effect on sustainable growth but not on actual growth. These findings contribute to the body of knowledge of financial management specifically in terms of determining dividend and financing policy, operational activities and bridging conflicting objectives of managers and shareholders. Furthermore, these findings have implications for the practice, especially for shareholders, in how to maintain and set sustainable growth targets in conditions of various risks in banking. For banks within the framework of ASEAN integration, it is important to place SGR as a measure of sustainable finance.

**Keywords:** bank industry; sustainable growth rate; risk

## 1. Introduction

Countries gathered within the framework of the ASEAN Economic Community (AEC, hereafter) agreed to establish the ASEAN Banking Integration Framework (ABIF, hereafter) in 2014. The main objective of ABIF is to prepare market access and freedom of banking operations in ASEAN member countries (Indonesia, Malaysia, the Philippines, Singapore, Thailand, and Vietnam), in the context of creating Qualified ASEAN Banks (QAB) [1]. However, banking integration in the ASEAN region brings new challenges and risks related to money market uncertainties and capital flow volatility [2,3]. Increasing global financial liberalization is adding to the severity of systemic banking risks among ASEAN countries [4,5]. The biggest risk during the 2007–2008 global crisis in the banking sector experienced by various ASEAN countries was liquidity problems caused by high credit risk with non-performing loans [5,6].

The dynamic and rapidly changing global financial environment creates various risks for the banking sector. The complexity of banking transactions puts pressure on efforts to increase competitiveness, carry out efficiency, improve the soundness of capital-based banks, and manage risk [4,7]. Therefore, the sustainability of banks in the Association of Southeast Asian Nations (ASEAN) depends on how efficiently and effectively risk is managed. The challenge for the banking industry is efforts to minimize risk and increase revenue, as a basic concept in the financial literature. The financial literature has explained high risk and high return, but risks that are too high can cause corporate bankruptcy or

bankruptcy, for example, high financial risk, besides that high credit risk and liquidity risk can disrupt banking integration and stability in ASEAN countries.

The measurement of risk for ASEAN countries is generally macro in nature, whereas basically banking performance on a micro basis contributes to the sustainability of an integrated banking sector. The main banking performance indicators besides the company's profitability performance are also the creation of value for shareholders. One important measurement of the performance of companies and shareholders is the Sustainable Growth Rate (SGR). SGR becomes an alternative for planning, evaluating, and controlling performance as well as controlling growth for the banking industry. Banks with high growth do not guarantee a good level of soundness, and banks with low growth can have an impact on losses and financial consequences. The Sustainable Growth Rate (SGR) is the sustainable growth rate of a company without financial difficulties. Determination of SGR is very important for companies for two reasons: first as a measurement of company performance and second as a means of controlling shareholders to control their equity.

An important concept offered is SGR as one of the essential parameters in corporate financial control, which places how the company can achieve maximum growth to increase revenue without adding new equity but still maintain its capital structure [8] forts to maintain business continuity and ensure value for shareholders. For this reason, shareholders need essential information to control their equity. It is not uncommon for companies with high sales growth to experience pressures in financial problems such as high debt levels, low asset utilization, and high costs, which impact financial losses and potential bankruptcy [9,10]. It is called "Grow and Broke" [8,11,12]. Therefore, shareholders need to control the financial structure through financial policies such as dividend policy and financial policy (source of funding) to control the company's sustainability in the long term.

Various studies have shown that the sustainable growth rate model undergoes various adjustments, confirmed by various empirical studies in various sectors. However, not many corporate performance appraisal practices place sustainable growth as an essential indicator for controlling company finances by shareholders. Most findings for the benefit of shareholders are still oriented to the size of earnings per share or earnings growth, whereas [13] had introduced the concept of SGR, which showed the difference between earning growth and SGR as "supportable growth" [14]. Furthermore, many researchers use the same indicators to determine SGR as a control tool for shareholders, and distinguish between actual growth rate, internal growth, and sustainable growth rate [8,15–17].

From several empirical studies and literacy, SGR has a relationship with risk, for example deviation of actual growth [18]; stochastic growth rate and dividend per share [19]; risk preference [20]; tax rate [21]; inflation [15]; liquidity risk [22]; financial innovation [23]; bankruptcy [10]; SGR and financial distress [16]; growth cycle stage [24]; and recession [25]. Ref. [25] distinguish the effect of recession on SGR in non-agricultural banks and agricultural banks where agricultural banks show more aggressiveness in producing an actual growth rate than SGR compared to non-agricultural banks. From various studies, it has not been detected how risk sensitivity influences SGR.

The banking industry is vulnerable to risk because its business process is to collect funds and distribute them to the public in the form of loans or as financial intermediaries. A bank will face risks, especially from loans. The risk that may occur is the default risk, also called default probability: the probability that the borrower will fail to make full and timely principal and interest payments, by the terms. Credit risk is the risk that the borrower will not repay the loan. Some literature and research findings indicate that credit risk can be measured by the Capital Adequacy Ratio (CAR) and Nonperforming Loan (NPL). Low CAR and NPL will affect cash flow, so it has a potential default risk, which can be measured by free cash flow [26–28].

In addition to these two risks, the potential risk that will affect revenue is operational risk. Operational risk is a risk that can be caused by internal factors, which can be measured by the level of efficiency ratio [29–31]. The higher the efficiency level, the higher the profit potential ratio will be, which will affect internal funding sources, which in turn can affect

the SGR level. Cost income ratio, cost-to-average asset ratio, or Risk-Weighted Assets (RWA) are used to determine the minimum amount of capital that banks and other financial institutions must have to reduce the risk of bankruptcy [32].

Many factors that can influence SGR have not been explored extensively, and several researchers have demonstrated empirically that there are significant differences between actual growth and SGR [18,33–36]. Most studies show that the most influential factor on SGR is profitability as an endogenous variable which can be caused by working capital factors [37], technological innovation [38], and intangible assets [39], which in turn will determine the model's dividend policy, where SGR will change according to the dividend payout ratio [19].

Many studies show a negative relationship between the level of debt and profitability, but the opposite is true for sales growth and profitability. On the other hand [23,40] in their studies on banks listed in Nifty 50 states that Return on Assets to Sales (ROA) has a negative impact on the SGR. Several studies reveal some factors that affect sales growth, such as innovation [41]; capital structure [42]; and sales professional [43]. However, bank SGR growth was found to be positively related to profit margins. The company can survive in the long term and be able to compete in the market only if the SGR trend is increasing. Therefore, any company introducing any product or service should prioritize SGR by introducing innovative products. Other researchers [44] stated that many previous studies around the world focused on bank performance and confirmed that there was a linear relationship between Non-IR and SGR in increasing profitability and reducing risk. Another study states that a negative relationship and non-interest income can increase operational risk. Ref. [45] concluded that although Non-IR has explanatory power for SGR and can maintain bank growth rate, the direct effect is not statistically significant, because there is a non-linear relationship between non-interest income and SGR, and bank size plays an important role to expand bank capacity in diversifying their income to maintain growth rates. Therefore, this study confirms the results of the study by [46] on nonlinear relationships and the effect sizes [47], and the bank size effect on risk [48].

The company retains profits to increase sales growth, meaning that it is intended for short-term goals and operational reasons. Although companies are using retained earnings for long-term investment purposes [49,50] in addition to using external funding sources, the SGR concept assumes that external funding sources are relatively fixed, so the company can show sales growth or increasing new markets developed from internal funding sources [8,15]. This is certainly related to the company's dividend policy. Some researchers show that the dividend policy will affect firm value and managerial motivation [51–53].

However, several studies show a contradiction between sales growth and funding sources, which shows that the higher the level of sales, the higher the SGR level, indicating the greater the need for funding and the higher the level of debt [22,54,55] or several other studies show that there is a relationship between NPM and leverage [19,56]. This draws attention to the development of the predictive model of how risk sensitivity is to SGR in the banking sector in several ASEAN countries.

Recently, the banking industry is facing various challenges, especially in competition. Various regulations for tightening banking regulations are currently an interesting issue. In order for banks to be able to compete, banks need to implement a diversification strategy to maximize total revenue and reduce risk with non-interest products/services. The non-interest products/services are expected to generate more income. Income diversification in the banking industry would be able to reduce income volatility and risk and will be better than merely defending traditional interest income [22,23,48]. Therefore, SGR should also be viewed as a way to reduce bank business risk.

The study of SGR is very important for the banking industry because SGR is a measure for long-term goals in determining revenue growth targets without financial difficulties. The banking industry is an industry that is prone to volatility, as Higgins shows that an important factor in determining SGR is inflation [15,16]. The banking industry as an industry plays an important role in overcoming various economic recoveries, so the

determination of SGR as important sustainability indicators pays attention to the various risks faced. This is in line with [57] who wrote that for sustainable growth, it is necessary to pay attention to risk management in the banking industry because inflation and a decrease in interest rates will increase market risk as factors influencing sustainable growth. A risk analysis of SGR was also carried out by [58] who explained the practice of risk management between Fill-Fledged Islamic banks and Islamic banks from conventional bank subsidiaries in Malaysia towards sustainable growth. His research documented that the Risk Weighted Capital Ratio (RWCR) of Islamic bank subsidiaries of conventional banks is higher than that of Full-Fledged Islamic banks. The determination of SGR is also very important when a bank makes an acquisition or merger, as stated by [59] in a study of bank mergers in the United State because SGR is a significant measurement of the performance of mergers and acquisitions in the long term.

Several studies of SGR in the banking industry, shown by [60] on the banking industry in Saudi, prove that margin profitability, retained earnings, asset turnover, and financial leverage have a positive effect on SGR and [61] using the SGR model on banks in Greece proved the same thing. However, it is different from [42], which proves that poor liquidity and asset quality have a negative effect on SGR, in a study of banks in Indonesia that are listed on the Indonesia Stock Exchange. This is in line with research [22] which proves that Loan Funding Ratio (LFR) and Non-Performing Loans (NPL) have a negative effect on SGR in studies of 22 banking industries in Indonesia of both private banks and government banks. NP is an indicator for measuring asset quality: the lower the NPL, the higher the SGR, while the LFR is an indicator of the liquidity ratio at a bank. The relationship between risk to SGR is also explained by [25], who states that the decision to determine SGR needs to be distinguished between agricultural and non-agricultural banks, as an example of the impact of the 2008 recession, in America. Ref. [25] proves that changes in margins have a positive impact on SGR at non-agricultural banks, whereas negative effects are found at non-agricultural banks. Another study was conducted at small and medium banks in Kilimanjaro, Tanzania, which proved that LFR, NPL, and BOPO had a negative effect on SGR [62]. This implies that bank risk has a negative effect on SGR. Besides that, the capital adequacy ratio determines SGR, because the growth rate of assets is supported by internal and external capital sources with the prerequisite capital adequacy ratio not changing, and the riskier assets being higher. The capital adequacy ratio must pay attention to risky assets to pursue growth [63].

Banking integration in the association of ASEAN countries requires a balanced performance among its members to compete globally. Therefore it is important to explain how the risks and levels of sustainable growth are in the banking industry in ASEAN countries because this can disrupt the soundness of banks and have an impact on economic growth in the ASEAN region. Besides that, this research will prove the impact of risks (business risk, operational risk, liquidity risk, and financial risk) on sustainable growth rates. These mentioned risks are important risks in the banking world as well as a measure of performance and corporate value.

## 2. Literature Review

### 2.1. Sustainable Growth Rate Model

Ref. [13] introduced the concept of sustainable growth rate which is the same as the concept of supportable growth [14] as an analytical tool for the benefit of long-term investors. Furthermore, ref. [64] introduced the concept of Sustainable growth rate (SGR) as a measure of company performance based on internal funding sources to achieve maximum sales growth at a certain level of profitability. Then, improvements and modifications are made to the SGR measurement [15–17,19,65–67]. Ref. [8] used a simple model with the formula sustainable growth rate = ROE × b, where b is the retention rate or (1-the payout rate). Another relatively different model was proposed by [12] by offering two models: the first is called the steady-state model using a balance sheet and performance ratio, assuming no new equity funding, and the source of funding is from retained earnings. Second,

the dynamic model uses a target ratio, namely (a) retention ratio, (b) Net Profit Margin, (c) Equity multiplier, (d) assets-to-sales ratio. The advantage of the Van Horne model is that it explains the occurrence of changes in assumptions or the environment. The assumption is that an increase in assets as the use of funds must equal an increase in liabilities and shareholder equity as a source of funding.

Ref. [64] proposed a sustainable growth model with a discrete time frame model and further developed with a sustainable time frame [15]. Higgin's SGR model consists of four accounting ratios, namely: dividend payments, profit margins, asset turnover, and capital structure. In the [15] model it is assumed that the company does not use new equity and the portion of retained earnings and additional debt is used for investment in assets, while the [16] suggested an SGR model which was relatively different from [64] model. The difference is in the Asset to Sales, and a continuous model of the capital structure ratio calculated at the beginning of the period. Ross places the issue of sustainable growth rate on the need for financial policy and growth in the long term. Ross stated that the sustainable growth rate is the maximum growth that can be achieved without additional external equity funding and still maintains a constant debt to equity ratio. Ross' SGR model shows that the sustainable growth rate is determined by four important variables, namely (1) profit margin, (2) dividend policy, (3) financial policy and (4) total asset turnover. An important issue raised in Van Horne's model is that an increase in assets as a use of funds must equal an increase in liabilities and shareholder equity as a source of funds [17]. Van Horne put forward two models, the first in the steady model and the dynamic model. The Steady model is formulated relatively differently from Higgins's model. Furthermore, Van Horne proposes SGR under conditions of changing assumptions. This model shows that the previous year's sales and equity at the end of the previous year serve as the foundation on which to build a model from year to year.

In the banking sector, the implementation of the Van Horne model is adjusted to the banking account, and in this case the difference is only in the sales account which is replaced by operating revenue (OR) and Profit Margin using the ratio of Net Operating Income to Operating Revenue.

### 2.2. SGR and Banking Risk Variability

Various SGR models and various empirical studies have been shown previously which still require extensive exploration of explanation. The essence of the initial concept of SGR was built on the importance of pricing earnings per share which responds to the dynamics of business development, through the retention ratio as an internal funding source to reflect value for shareholders. The most crucial risk is a financial risk that can be caused by global companies and institutions. The financial risk order, in general, includes elements of currency risk, interest risk, and commodity risk [68–70].

In the banking industry, which is strict with various regulations, the BASEL committee has been appointed as a supervisor, especially in terms of risk control in the banking sector. Various types of risks that can occur include operational risk, business risk, liquidity risk, credit risk, market risk, reputational risk, currency risk, and others [71–73]. The BASEL Committee also sets a minimum standard of adequacy to cover various banking risks. This study only discusses some important risks.

Operational risk is defined as the risk of loss resulting from the inability or failure of internal processes, people, and systems or various external events [74]. One indicator of operational risk can be seen from the efficiency ratio, including cost to income and cost to assets ratio. Cost to income ratio is the ratio between operating costs and operating income. The lower the ratio of costs to revenues, the better the company's performance. Likewise, the lower the ratio, the more efficiencies the company can achieve in the period [75,76]. Business risk is a risk that arises from a long-term strategy, in this case, the bank cannot defend the business from the dynamics of competition. Business risk is the exposure a company or organization has to factor that will reduce its profits or cause it to fail. Anything

that threatens the company's ability to achieve its financial goals is considered a business risk [77–79].

Liquidity risk is the risk that arises from the company's inability to meet its immediately maturing obligations [25]. Liquidity risk can be seen from the level of Loan to Deposit Ratio. Liquidity risk occurs when individual investors, businesses, or financial institutions are unable to meet their short-term debt obligations [80]. An investor or entity may not be able to convert an asset into cash without surrendering capital and income due to a lack of buyers or an inefficient market. Basel determines liquidity risk by comparing the liabilities and liquid assets listed in the company's financial statements. Basel determines the size of liquidity with the Liquidity Coverage Ratio (LCR). LCR is a requirement under Basel III where banks are required to have high- quality liquid assets sufficient to fund cash outflows for 30 days. Another measurement is LDR: Loan-to-deposit ratio is used to assess bank liquidity by comparing total bank loans with total deposits for the same period.

Financial risk is the possibility of losing money on an investment or business venture. Some of the more common and distinct financial risks include credit risk, liquidity risk, and operational risk. Financial risk is a type of hazard that can result in loss of capital to interested parties. Various literature places financial risk as robust risk, as a reflection of uncertainty, which can be quantitatively measured by an error model at a certain probability level [81]. The concept of robust risk is often used in the measurement of probability which is different from the conditional measurement with the level of variance or standard deviation. The measurements using Conditional Value At Risk (CVaR) are often discussed in risk management [82,83]. One of the risk measurement instruments is VaR (Value at Risk). VaR can be interpreted as an estimate of the maximum potential loss in a certain period with a certain confidence level and under normal market conditions. In this study, we use Risk Weighted Assets as an important indicator in assessing bank performance. Capital requirements are based on a risk assessment for each type of bank asset [84].

Risk sensitivity to SGR is very crucial to bridge the relationship between shareholders and managers, in terms of evaluating company performance. Therefore, the focus of this research, apart from describing several SGR measurement models and comparing them with actual growth, also predicts how sensitive various important risks are to SGR in the banking sector of ASEAN.

## 3. Research Design

### 3.1. Sample

The focus of the research is the commercial banking sector in various ASEAN countries. Banking data that can be accessed for each country is summarized as follows: (1) Indonesia, 36 Banks; (2) Malaysia, 15 Banks; (3) Philippines, 9 Banks; (4) Singapore, 3 Banks; (5) Thailand, 6 Banks. The sample data consist of 69 banks for five years from 2015 to 2019, meaning that all n-data panels are 330 samples. This sample size is considered sufficient to be statistically generalized to the banking population in ASEAN countries. The data source is Bank Focus Data Base.

### 3.2. Model Specification

This study was to examine the effect of risk on SGR on panel data using multiple regression analysis techniques. To produce a good regression model, a model specification test is needed. Selection of the best model is carried out by (1) the Chow test, this test is used as a method for selecting a model in panel data regression, namely between the fixed effect model and the pooled regression model. If the F value < 0.05, then the fixed effect model is better than the common effect. (2) The Hausman Test was conducted to compare which model is the most appropriate between Fixed Effects and Random Effects. If the random cross-section probability value is <0.05, it can be concluded that the Fixed Effect model is more appropriate than the Random Effect model.

The Regression model is as follows:

$$VH'SSGRB_{it} = \propto_{it} + \delta_{1it}COSTINCRATIO + \delta_{2it}COASSRATIO + \delta_{3it}REV_{RISK} + \delta_{4it}LDR + \delta_{5it}CAR + \delta_{6it}EQRWA + \delta_{7it}RWAI + \delta_{8it}GASSETS + \cup_{it}$$

(1)

$$ACT\_GR_{it} = \mu_{it} + \varphi_{1it}COSTINCRATIO + \varphi_{2it}COASSRATIO + \varphi_{3it}REV_{RISK} + \varphi_{4it}LDR + \varphi_{5it}CAR + \varphi_{6it}EQRWA + \varphi_{7it}RWAI + \varphi_{8it}GASSETS + \in_{it}$$

(2)

**Measurement and Definition of Variable**

In this study, there are three SGR measurement models because several studies have shown there are differences in SGR with different measurements. The SGR measurement model uses (1) Higgin's SGR, (2) Ross's SGR and Van Horne's SGR shown below.

Higgin's SGR discrete model is as follows:

$$HG\prime S\_SGR = RR \times NPM \times ATO \times FL$$

(3)

where *RR* is the profit retention ratio calculated as retained earnings divided by net income, *NPM* is the net profit margin calculated as net income divided by sales, *ATO* is asset turnover calculated as sales divided by total assets, and FL is financial leverage calculated as total assets divided by book equity.

Ross' SGR model shows that the sustainable growth rate is determined by four important variables, namely (1) profit margin, (2) dividend policy, (3) financial policy, and (4) total asset turnover. Ross's model is as follows (Ross et al. 1996:92):

$$ROS\prime S\_SGR = \frac{b \times ROE}{1 - (b \times ROE)}$$

(4)

*ROE* = Return on equity; *ROE* = Profit margin × Total Asset Turnover × Equity Multiplier = Net Income/Total Equity

*b* = Plowback (retention) ratio = Addition to retained earnings/Net Income Actual Growth, Internal Growth Sustainable Growth in several ASEAN countries

Van Horne put forward two models, the first in the steady model and the dynamic model. The steady model is formulated relatively differently from Higgins's model. Van Horne's SGR steady model is formulated as follows:

$$VH\prime S\_GR_B = \frac{RR\left(\frac{NOI}{OR}\right)\left(1 + \frac{TL}{Eq}\right)}{\left(\frac{TA}{OR}\right) - \left[RR\left(\frac{NOI}{OR}\right)\left(1 + \frac{TL}{Eq}\right)\right]}$$

(5)

SGR is computed using operating revenue data, retention ratio, equity multiplier return on equity, and financial leverage. For the risk variable, the concept of risk in the banking sector is used, which consists of business risk, operational risk, liquidity risk, and financial risk. The definition of each variable can be explained in Table 1.

**Table 1.** Operational Definition of Variables.

| No | Variable | Proxy | Definition | Indicator |
|---|---|---|---|---|
| 1. | Higgins' SGR Model (1991) HG'S_SGR | Higgins' SGR model consists of four accounting ratios, namely: dividend payments, profit margins, asset turnover, and capital structure | Higgins proposes the SGR model as a revenue/sales target by using internal funding sources as the impact of asset turnover and profit margin on a fixed leverage financial condition | $HG\prime S\_SGR = RR \times NPM \times ATO \times FL$ |

**Table 1.** *Cont.*

| No | Variable | Proxy | Definition | Indicator |
|---|---|---|---|---|
| 2. | Ross's SGR Model ROS'S_SGR | Using a simpler model consisting of the variable retained earnings as the plowback ratio and ROE | A model that explains SGR targets with internal funding sources through dividend policy and value for shareholders through Return to Equity | $ROS\prime S\_SGR = \frac{b \times ROE}{1-(b \times ROE)}$ |
| 3. | This Research's SGR Model VH'S_SGRB | Consists of Retention ratio, Net Operating Income, Operating Revenue, Ratio of Total Liability to Equity, and Ratio of Total Assets to Operating Revenue. | This model uses the Van Horne SGR model, which is implemented in banking accounts (in this study), namely the target for banking operating income growth that can be achieved with relatively fixed external sources of funds. | $VH\prime S\_GR_B = \frac{RR\left(\frac{NOI}{OR}\right)\left(1+\frac{TL}{Eq}\right)}{\left(\frac{TA}{OR}\right) - \left[RR\left(\frac{NOI}{OR}\right)\left(1+\frac{TL}{Eq}\right)\right]}$ |
| 4. | Internal Growth I_GR | Internal growth rate According to Ross, it includes Return On Assets and Retention Ratio variables | Is an indicator of company growth that guarantees the company's operations and income through the rate of return on assets | $I\_GR = \frac{ROA \times b}{1-ROA \times b}$ |
| 5. | Actual Growth ACT_GR | Growth of banking operating revenue. | Revenue growth is the increase, or decrease, in a company's operating revenue between two periods. | $ACT\_GR$ $= \frac{Operating\ Revenue_t - Operating\ Revenue_{t-1}}{Operating\ Revenue_{t-1}}$ |
| 6. | REV_RISK | Business Risk | Business risk is the exposure a company or organization has to factor that will lower its profits or operating revenue (OpRev) or lead it to fail. Anything that threatens a company's ability to achieve its financial goals is considered a business risk. In this case, it is measured by the standard deviation of the volatility of operating revenue within a period of 5 years | $Std_{OpRev} = \sqrt{\frac{\sum_{i=1}^{j=5}\left(X_{ij}-\overline{X}_{ij}\right)^2}{n-1}}$ X = operating revenue |
| 7. | COSTINCRATIO | Operational Risk Operating Cost To Operating Income Ratio | The cost to operating income ratio is one of the efficiency ratios used to gauge an organization's efficiency. It is used to compare the operating expenses of a bank vis-à-vis its income. The lower the cost to income ratio the better the company's performance. It depicts the efficiency at which the bank is being run. | $COSTINCRATIO = \frac{operating\ cost}{Operating\ income}$ |
| 8. | COASSRATIO | Cost To Asset Ratio | Cost to Assets Ratio (%) is an efficiency ratio that measures the operating expenses, i.e., non-interest expenses, of a bank about its size or the asset base | $COASSRATIO = \frac{operating\ cost}{Total\ Assets}$ |
| 9. | LDR | Liquidity Risk is the risk that occurs if the company is unable to fulfill its obligations immediately in the short term which can be proxied by Loan To Deposit Ratio (LDR) and Non-Performing Loan (NPL) | The loan-to-deposit ratio is used to assess a bank's liquidity by comparing a bank's total loans to its total deposits for the same period. To calculate the loan to-deposit-ratio, divide a bank's total amount of loans by the total amount of deposits for the same period. | $LDR = \frac{Total\ Loans}{Total\ Deposits}$ |

**Table 1.** *Cont.*

| No | Variable | Proxy | Definition | Indicator |
|---|---|---|---|---|
| 10. | NPL | Liquidity Risk | A non-performing loan (NPL) is a loan in which the borrower defaults and does not make scheduled principal or interest payments for some time | The total NPL is divided by the total number of loans in the bank's portfolio. The ratio can also be expressed as a percentage of the bank's non-performing loans. |
| 11. | Eq to RWA | Financial Risk | The equity-to-risk weighted assets ratio (WRA) will help determine whether or not a bank has enough equity to take on any losses before becoming insolvent and losing depositor funds. It's important for a bank to monitor this ratio and adhere to regulatory requirements to avoid going insolvent and to protect its clients and the larger economy as a whole. | $EQRWA = \frac{Total\ Equity}{RWA}$ |
| 12. | RWAI | Financial Risk | Risk weighted asset intensity (RWA / Total Assets). Weighted assets, or RWA, are used to link the minimum amount of capital that banks must have, with the risk profile of the bank's lending activities (and other assets). The more risk a bank is taking, the more capital is needed to protect depositors | $RWAI = \frac{RWA}{Total\ Assets}$ |
| 13. | GASSETS | Growth assets | Growth assets are assets that generate a return both from capital growth and from the distribution of profits through retention ratio and external funding | $\frac{Total\ Assets_t - Total\ Assets_{t-1}}{Total\ Assets_{t-1}}$ |

## 4. Results and Discussion

### 4.1. Descriptive of SGR and AGR in Various ASEAN Countries

As mentioned before the data consist of 69 banks for the year 2015 to 2019 ASEAN, excluding Vietnam. Table 2 shows the variables descriptively for growth including actual growth rate, internal growth rate, and sustainable growth rate for the Higgins, Ross, and Van-Horne models. From the sample data, it can be confirmed that there is a very high difference between actual growth, internal growth, and SGR. For all countries in ASEAN (excluding Vietnam), the average actual growth for five years shows negative growth. This is also shown in the graph of actual growth which shows fluctuating revenue growth, and almost all countries experienced a decline in operating revenue for the banking sector. This is the impact of the COVID-19 pandemic which has disrupted macroeconomic activities. However, internal growth still shows a positive average value as an indication that operating revenue can still cover asset utilization and company operations. The SGR target policy for various countries, there is no very significant difference, with the SGR level being between 6 to 10 percent. Likewise, there is no significant difference between the three Higgins, Ross, and Van-Horne models ranging from 7 to 8 percent. This finding also indicates that the condition of the banking sector at a macro level for various countries is the same and systemic.

**Table 2.** Variables Actual Growth, Internal Growth, Sustainable Growth in several ASEAN countries.

| Country | | N | Minimum | Maximum | Mean | Std. Deviation |
|---|---|---|---|---|---|---|
| Indonesia | HG'S_SGR | 180 | −0.6390 | 0.2810 | 0.063933 | 0.1258829 |
| | ROS'S_SGR | 180 | −0.7230 | 0.3310 | 0.075556 | 0.1455922 |
| | VH'S_SGRB | 180 | −0.4200 | 0.4950 | 0.102494 | 0.1386616 |
| | I_GR | 180 | −0.1050 | 0.0455 | 0.008889 | 0.0166827 |
| | ACT_GR | 180 | −2.6377 | 0.7152 | −0.108549 | 0.2942697 |
| Malaysia | HG'S_SGR | 60 | 0.0000 | 0.1440 | 0.071383 | 0.0308007 |
| | ROS'S_SGR | 60 | 0.0000 | 0.1560 | 0.080000 | 0.0324695 |
| | VH'S_SGRB | 60 | 0.0000 | 0.1850 | 0.088300 | 0.0380937 |
| | I_GR | 60 | 0.0000 | 0.0118 | 0.006562 | 0.0023930 |
| | ACT_GR | 60 | −0.3126 | 0.2882 | −0.008435 | 0.1171124 |
| Philippines | HG'S_SGR | 45 | 0.0320 | 0.1520 | 0.080111 | 0.0290002 |
| | ROS'S_SGR | 45 | 0.0380 | 0.1730 | 0.091511 | 0.0320650 |
| | VH'S_SGRB | 45 | 0.0400 | 0.2100 | 0.102178 | 0.0399080 |
| | I_GR | 45 | 0.0027 | 0.0164 | 0.009191 | 0.0031024 |
| | ACT_GR | 45 | −0.4826 | 0.0949 | −0.099329 | 0.1394821 |
| Singapore | HG'S_SGR | 15 | 0.0150 | 0.1650 | 0.080133 | 0.0404031 |
| | ROS'S_SGR | 15 | 0.0170 | 0.1760 | 0.087467 | 0.0422947 |
| | VH'S_SGRB | 15 | 0.0170 | 0.2130 | 0.098200 | 0.0535300 |
| | I_GR | 15 | 0.0011 | 0.0091 | 0.006147 | 0.0019504 |
| | ACT_GR | 15 | −0.4608 | 0.0504 | −0.104107 | 0.1331989 |
| Thailand | HG'S_SGR | 28 | −0.5210 | 0.1570 | 0.069107 | 0.1224979 |
| | ROS'S_SGR | 28 | −0.6450 | 0.1770 | 0.076393 | 0.1484489 |
| | VH'S_SGRB | 28 | −0.3920 | 0.2150 | 0.099536 | 0.1106414 |
| | I_GR | 28 | −0.0906 | 0.0166 | 0.006271 | 0.0195608 |
| | ACT_GR | 28 | −0.3023 | 0.5287 | −0.045150 | 0.1544876 |

Figure 1 shows graphically the development of actual growth, internal growth, and sustainable growth for 5 years (2015–2019) in the banking sector in various ASEAN countries. Table 2 shows a relatively similar pattern for several countries within ASEAN, almost all countries in the banking sector showed a declining growth rate, and Thailand showed a drastic decline in 2019.

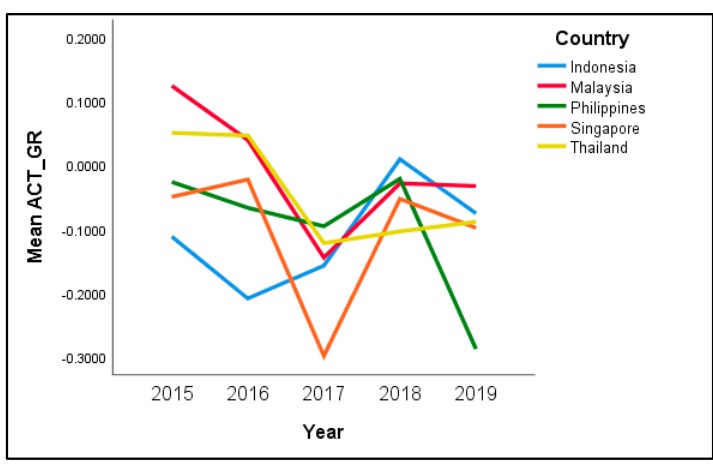
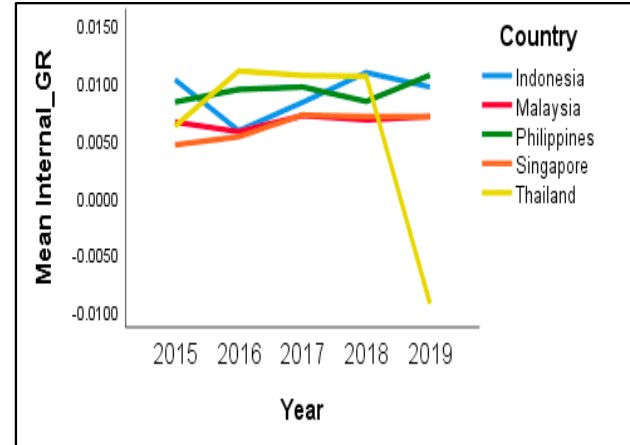

**Figure 1.** *Cont.*

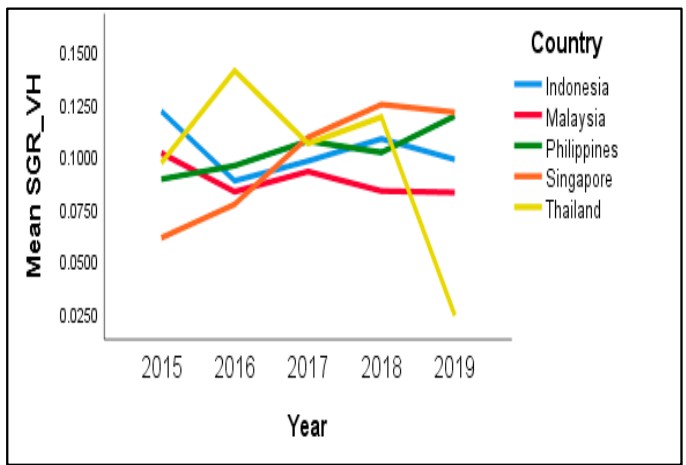 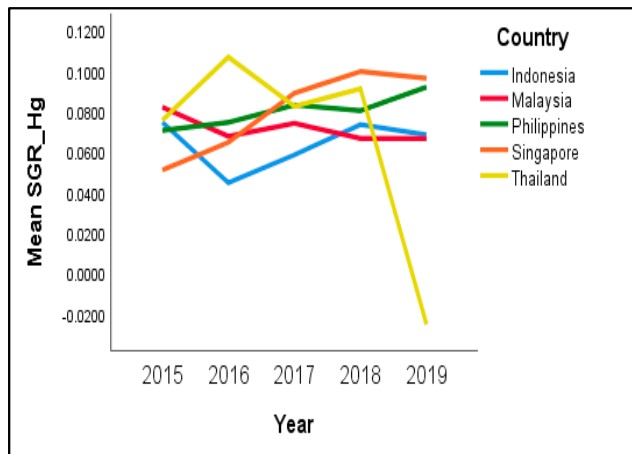

**Figure 1.** Actual Growth rate, Internal Growth, Van Horne's SGR model, and Higgins' Model from 2015 to 2019 in several ASEAN countries.

*4.2. Test Differences between SGR and AGR among ASEAN Countries*

The comparison among countries of the relationship between actual growth, internal growth, and sustainable growth is shown in Table 3. Paired sample *t*-test between the three variables in almost all countries showed a very significant difference, except for Malaysia and Thailand which did not show any difference between the internal growth variables and actual growth. This is an indication that the operating revenue growth rate is in line with the dividend policy and the company's asset utilization. The difference between internal growth and actual growth as an indication that the company achieves actual growth is not followed by internal growth in terms of asset utilization and dividend policy or retention ratio. This can happen because of the dominance of external funding sources compared to internal funding. Therefore, if this happens, it is necessary to balance growth (balance growth), through restructuring funding sources by increasing retained earnings or efficient use of assets, in the banking industry of ASEAN countries. Furthermore, there is a significant difference between actual growth and sustainable growth, which shows that actual growth is much lower when compared to sustainable growth, as an indication that operating revenue growth has not guaranteed the importance of value for shareholders. SGR is the operating revenue target in the banking sector which is expected if the company uses internal funds through the retention ratio, with the expectation of increasing shareholder value. The low actual growth compared to SGR is an indication of not achieving the operating revenue target to ensure the sustainability of the company from the side of the shareholders. This can happen because the return on equity target is too large or the company has not worked optimally in generating revenue income. A detailed description of the difference in comparison between actual growth and internal growth and sustainable growth in the banking sector in ASEAN can be seen in Table 3.

**Table 3.** Paired Sample Test of Internal Growth, Actual Growth, and Sustainable Growth of the banking industry in several ASEAN countries.

| | Country | | Paired Differences | | | | | T | Sig. (2-tailed) |
| | | | Mean | Std. Deviation | Std. Error Mean | 95% Confidence Interval of the Difference | | | |
| | | | | | | Lower | Upper | | |
|---|---|---|---|---|---|---|---|---|---|
| Indonesia | Pair 1 | I_GR-ACT_GR | 0.11743 | 0.29530 | 0.02201 | 0.07400 | 0.16087 | 5.335 | 0.000 |
| | Pair 2 | VH'S_SGRB-ACT_GR | 0.21104 | 0.32895 | 0.02451 | 0.16266 | 0.25942 | 8.607 | 0.000 |
| | Pair 3 | I_GR-VH'S_SGRB | −0.09360 | 0.12374 | 0.00922 | −0.11180 | −0.07540 | −10.148 | 0.000 |

**Table 3.** *Cont.*

| | Country | | Paired Differences | | | | | T | Sig. (2-tailed) |
|---|---|---|---|---|---|---|---|---|---|
| | | | Mean | Std. Deviation | Std. Error Mean | 95% Confidence Interval of the Difference | | | |
| | | | | | | Lower | Upper | | |
| Malaysia | Pair 1 | I_GR-ACT_GR | 0.01499 | 0.11699 | 0.01510 | −0.01522 | 0.04522 | 0.993 | 0.325 |
| | Pair 2 | VH'S_SGRB-ACT_GR | 0.09673 | 0.12435 | 0.01605 | 0.06461 | 0.12885 | 6.026 | 0.000 |
| | Pair 3 | I_GR-VH'S_SGRB | −0.08173 | 0.03685 | 0.00475 | −0.09125 | −0.07221 | −17.179 | 0.000 |
| Philippines | Pair 1 | I_GR-ACT_GR | 0.10852 | 0.14059 | 0.02095 | 0.06628 | 0.15075 | 5.178 | 0.000 |
| | Pair 2 | VH'S_SGRB-ACT_GR | 0.20150 | 0.15743 | 0.02346 | 0.15420 | 0.24880 | 8.586 | 0.000 |
| | Pair 3 | I_GR-VH'S_SGRB | −0.09298 | 0.03736 | 0.00557 | −0.10421 | −0.08176 | −16.693 | 0.000 |
| Singapore | Pair 1 | I_GR-ACT_GR | 0.11025 | 0.13365 | 0.03450 | 0.03623 | 0.18426 | 3.195 | 0.006 |
| | Pair 2 | VH'S_SGRB-ACT_GR | 0.20230 | 0.15614 | 0.04031 | 0.11583 | 0.28877 | 5.018 | 0.000 |
| | Pair 3 | I_GR-VH'S_SGRB | −0.09205 | 0.05181 | 0.01337 | −0.12074 | −0.06335 | −6.881 | 0.000 |
| Thailand | Pair 1 | I_GR-ACT_GR | 0.05142 | 0.16013 | 0.03026 | −0.01067 | 0.11351 | 1.699 | 0.101 |
| | Pair 2 | VH'S_SGRB-ACT_GR | 0.14468 | 0.21187 | 0.04004 | 0.06252 | 0.22684 | 3.613 | 0.001 |
| | Pair 3 | I_GR-VH'S_SGRB | −0.09326 | 0.09232 | 0.01744 | −0.12906 | −0.05746 | −5.345 | 0.000 |

*4.3. Descriptive of Risk Variability in Various ASEAN Countries*

The results of statistical descriptions show different characteristics among the banking sectors of ASEAN. For example, the average business risk ranges from 11 to 17.7 percent, the highest business risk is in Thailand at 17.7 percent, followed by Indonesia with a risk level of 15.9 percent and the lowest business risk is in the Philippines with 11.2 percent followed by Malaysia at 11.78 percent.

Operational risk shows the risk that can occur due to internal failure, which in this case can be measured by the level of company efficiency. The level of asset-based efficiency can be measured by the ratio of costs to total assets. The lower the ratio of cost to total assets, the higher the level of efficiency. Likewise with the measurement of the level of efficiency with the ratio of costs to total income, the lower the ratio of total costs to total operating income, the more efficient. Table 4 shows that the highest level of efficiency based on assets is in Singapore, followed by Malaysia, while the lowest efficiency level is in Indonesia, followed by the Philippines. However, when viewed from the level of efficiency towards income, it shows that the most efficient country is Malaysia followed by Thailand, and the least efficient country is Indonesia. The lower the level of efficiency, the higher the operational risk in the banking industry.

**Table 4.** Descriptive Statistics of Independent Variable in Several ASEAN Countries.

| Country | | N | Minimum | Maximum | Mean | Std. Deviation |
|---|---|---|---|---|---|---|
| Indonesia | REV_RISK | 180 | 0.0367 | 1.7096 | 0.159716 | 0.1566431 |
| | COSTINCRATIO | 180 | 32.2400 | 442.4600 | 64.090833 | 36.7747682 |
| | COASSRATIO | 180 | 1.2400 | 6.4800 | 3.360833 | 1.0730445 |
| | LDR | 180 | 0.2119 | 0.6481 | 0.439227 | 0.0666696 |
| | NPL | 180 | 0.0500 | 23.9100 | 4.379056 | 3.8455255 |
| | EQRWA | 180 | 10.3300 | 49.0700 | 21.864722 | 6.7151925 |
| | RWAI | 180 | 41.8200 | 90.0400 | 71.163611 | 11.3619166 |
| | CAR | 180 | 12.5800 | 45.8500 | 21.438444 | 5.6695979 |
| | GASSETS | 180 | −29.2600 | 280.7400 | 12.703611 | 27.6811875 |

**Table 4.** *Cont.*

| Country | | N | Minimum | Maximum | Mean | Std. Deviation |
|---|---|---|---|---|---|---|
| | REV_RISK | 60 | 0.0297 | 0.4350 | 0.117872 | 0.0822158 |
| | COSTINCRATIO | 60 | 30.5400 | 63.9900 | 46.436833 | 7.0330444 |
| | COASSRATIO | 60 | 0.7400 | 2.6800 | 1.394167 | 0.4323264 |
| | LDR | 60 | 0.3432 | 1.7138 | 0.484962 | 0.2460794 |
| Malaysia | NPL | 60 | 0.4800 | 11.3100 | 2.251667 | 1.9705365 |
| | EQRWA | 60 | 12.4600 | 86.4200 | 19.897667 | 12.8547205 |
| | RWAI | 60 | 45.5900 | 79.7300 | 61.133833 | 8.5147746 |
| | CAR | 60 | 14.7200 | 86.7300 | 21.014167 | 12.5477603 |
| | GASSETS | 60 | −34.4100 | 16.3200 | 1.938667 | 8.9689163 |
| | REV_RISK | 45 | 0.0275 | 0.2136 | 0.112082 | 0.0455703 |
| | COSTINCRATIO | 45 | 45.5400 | 77.0300 | 62.385333 | 7.9186422 |
| | COASSRATIO | 45 | 1.7000 | 5.0000 | 2.970444 | 0.9004240 |
| | LDR | 45 | 0.2548 | 0.5388 | 0.371433 | 0.0694828 |
| Philippines | NPL | 45 | 0.4100 | 6.0500 | 2.780889 | 1.5011962 |
| | EQRWA | 45 | 12.8900 | 24.4900 | 16.840222 | 2.8241004 |
| | RWAI | 45 | 65.6500 | 91.7600 | 77.049556 | 6.9288832 |
| | CAR | 45 | 12.2100 | 24.3100 | 15.895333 | 2.5656220 |
| | GASSETS | 45 | −0.9700 | 34.8400 | 13.506444 | 8.4006180 |
| | REV_RISK | 15 | 0.0304 | 0.2366 | 0.131647 | 0.0639605 |
| | COSTINCRATIO | 15 | 42.9200 | 71.6600 | 51.659333 | 10.9255680 |
| | COASSRATIO | 15 | 0.9200 | 1.9300 | 1.278000 | 0.3699073 |
| | LDR | 15 | 0.3388 | 0.4279 | 0.399733 | 0.0221359 |
| Singapore | NPL | 15 | 0.0200 | 1.7800 | 0.974333 | 0.6872067 |
| | EQRWA | 15 | 15.0000 | 26.9700 | 19.598000 | 3.1763483 |
| | RWAI | 15 | 26.5300 | 63.5000 | 45.630000 | 12.3257495 |
| | CAR | 15 | 15.6000 | 22.5000 | 17.320000 | 1.6410798 |
| | GASSETS | 15 | −2.7500 | 54.3300 | 9.439333 | 13.2043340 |
| | REV_RISK | 28 | 0.0426 | 0.8552 | 0.177132 | 0.2177855 |
| | COSTINCRATIO | 28 | 35.6000 | 66.1600 | 49.193929 | 8.7420581 |
| | COASSRATIO | 28 | 1.4300 | 2.5400 | 2.032857 | 0.2433540 |
| | LDR | 28 | 0.1534 | 0.4812 | 0.410889 | 0.0885771 |
| Thailand | NPL | 28 | 2.1800 | 16.9200 | 4.478214 | 2.7813081 |
| | EQRWA | 28 | 12.1900 | 47.2900 | 20.048929 | 8.0630820 |
| | RWAI | 28 | 54.2700 | 75.3200 | 66.805714 | 5.8319017 |
| | CAR | 28 | 14.8500 | 43.6900 | 20.778214 | 6.6480084 |
| | GASSETS | 28 | −18.5500 | 14.4600 | 1.759286 | 7.2854105 |

Liquidity risk is measured by LDR and NPL. A higher LDR ratio indicates the bank does not have enough liquidity to cover unexpected funding needs. Liquidity, or the ability to fund increased assets and meet liabilities as they come due, is critical to the survival of any banking organization. Healthy liquidity can reduce the likelihood of serious problems [85]. For this reason, liquidity analysis requires bank management not only to measure the bank's liquidity position on an ongoing basis but also to examine how funding requirements will develop in various scenarios, including adverse conditions. Liquidity oversight by the BASEL committee is currently focused on developing a greater understanding of how banks manage their liquidity globally, consolidating including in terms of technological and financial innovation, as a new way to fund bank activities in liquidity management. Table 3 shows the lowest LDR ratio is in the Philippines while the highest LDR is in Malaysia. This means that the highest level of liquidity risk is in Malaysia, followed by Indonesia, Thailand, Singapore, and the Philippines. However, in terms of NPL means the other fails to pay and does not make scheduled principal or interest payments for some time. The higher the NPL, the higher the liquidity risk. The country with the highest liquidity risk in terms of NPL is Thailand, followed by Indonesia, while the country with the lowest NPL is Singapore.

Financial risk can be seen from the ratio of equity-weighted assets to risk which will help determine whether the bank has sufficient equity to cover losses before becoming bankrupt and losing depositors' funds. The intensity of risk-weighted assets (RWA/Total Assets) in this case is given the notation RWAI, used to relate the minimum amount of capital that must be owned by a bank, with the risk profile of the bank's credit activities (and other assets). The greater the risk a bank takes, the more capital it needs to protect depositors. In this study, financial risk is given the notation EQRWA, namely the ratio of equity to financial risk. The higher EQRWA means the lower the bank's financial risk, as well as RWAI, namely the risk that must be borne by the bank based on the number of assets. The data shows that the highest financial risk is in the Philippines followed by Indonesia, and Malaysia, and the lowest financial risk is in Singapore. However, all Banks have complied with regulatory requirements for a capital adequacy ratio above 8 percent. This is in line with the BASEL II agreement issued in 2004, recommending having a total capital of at least 8 percent of risk-weighted assets as measured by the Capital Adequacy Ratio (CAR).

### 4.4. Effect of Risk on Sustainable Growth Rate (SGR) and Actual Growth (ACT_GR)

This section will explain how (1) these risks have an impact on the sustainable growth rate as a banking operating revenue target as a reflection of the interests of the banking sector shareholders through dividend policy; (2) how the effect of risk on the actual growth of operating revenue is as a reflection of the company's managerial interests. In this study, the SGR used is the Van-Horne model and actual growth as operating revenue growth. The average SGR of the Van Horne model shows a higher SGR level than the other models mentioned above. For testing, the influence of risk on SGR panel regression is employed. From the test, it concludes that the fixed effect model is more appropriate than the pool regression model. Furthermore, the Hausman Test is conducted to compare the fixed effect with the random effect, it reveals that the Fixed Effect model is more appropriate than the random effect model. Based on these two tests, it can be concluded that the fixed effect model is better than the random effect model. The fixed effect model above is free from violations of the classical assumption test. Based on the model accuracy test, it can be shown that the fixed cross-section model shows an R-Square value of 87.5 percent for the weighted statistic model and 78.6 percent, meaning that variations in all risk variables can explain changes in SGR. The following Table 5 shows the specification test for the panel data regression.

**Table 5.** Effects Specification Model: The Effect of Various Risks on VH'S_SGRB and ACT_GR.

| Cross-Section Fixed (Dummy Variables) | Variable | |
|---|---|---|
| | **VH'S_SGRB** | **ACT_GR** |
| Weighted Statistics | | |
| R-squared | 0.646174 | 0.625470 |
| Adjusted R-squared | 0.542856 | 0.516107 |
| S.E. of regression | 0.072336 | 0.165748 |
| F-statistic | 6.254264 | 5.719214 |
| Prob(F-statistic) | 0.000000 | 0.000000 |
| Mean dependent var | 0.102265 | −0.085289 |
| S.D. dependent var | 0.106987 | 0.238272 |
| Sum squared resid | 1.308135 | 6.868103 |
| Durbin-Watson stat | 2.077502 | 2.333971 |

The F-statistic value in the regression results is 22.89085 with a probability value of $0.0000 < (0.05)$. This shows that the independent variables in this model (REV_RISK, COSTINCRATIO, COASSRATIO, LDR, NPL, EQRWA, RWAI, GASSETS) can explain changes in SGRVH. The coefficient value of $R^2$ square shows the result of 87.53%, which means that the independent variables can explain the effect on SGRVH of 87.53%.

The effect of risk on the actual growth of banking operating revenue as managerial ability in overcoming various risks. The model accuracy test was carried out by the F test, showing the F-statistical value in the regression results of 7.83 with a probability value of 0.0000 < (0.05). This shows that the independent variables in this model (REV_RISK, COSTINCRATIO, COASSRATIO, LDR, NPL, EQRWA, RWAI, GASSETS) can explain changes in ACT_GR as a reflection of actual growth. The coefficient value R2 square shows the result of 0.712285 or 71.23%, which means that the independent variables can explain their effect on SGRVH of 71.23%. The details can be seen in Table 6.

**Table 6.** Risk sensitivity to sustainable growth and actual growth in the banking industry sector in several ASEAN countries.

| Variable | | Dependent Variable | | | | | |
|---|---|---|---|---|---|---|---|
| | | VH'S_SGRB | | | ACT_GR | | |
| | Proxy | Coefficient | Std. Error | Prob. | Coefficient | Std. Error | Prob. |
| C | | 0.358978 | 0.052247 | 0.0000 | −0.060437 | 0.119716 | 0.6141 |
| COSTINCRATIO | Operational Risk | −0.001565 | 0.000277 | 0.0000 | 0.002464 | 0.000634 | 0.0001 |
| COASSRATIO | Operational Risk | 0.020388 | 0.006805 | 0.0030 | −0.036263 | 0.015593 | 0.0208 |
| REV_RISK | Business Risk | 0.208373 | 0.055755 | 0.0002 | −0.336504 | 0.127753 | 0.0090 |
| LDR | Liquidity Risk | 0.094102 | 0.046384 | 0.0435 | 0.058926 | 0.106283 | 0.5798 |
| NPL | Liquidity Risk | −0.009763 | 0.001868 | 0.0000 | 0.001188 | 0.004280 | 0.7817 |
| CAR | Financial Risk | 0.000541 | 0.002557 | 0.8327 | 0.001650 | 0.005860 | 0.7785 |
| EQRWA | Financial Risk | −0.002385 | 0.002376 | 0.3163 | −0.001856 | 0.005444 | 0.7334 |
| RWAI | Financial Risk | −0.003189 | 0.000584 | 0.0000 | 0.000291 | 0.001339 | 0.8280 |
| GASSETS | Growth Assets | −0.000032 | 0.000234 | 0.8909 | −0.006747 | 0.000537 | 0.0000 |

Based on the above regression results (Table 6), the following model equation is obtained:

$$\text{VH'S\_SGRB} = 0.358978 - 0.001565 \text{ COSTINCRATIO (sig)} + 0.020388 \text{ COASSRATIO (sig)} + 0.208373 \text{ REV\_RISK (sig)} + 0.094102 \text{ LDR (sig)} - 0.009763 \text{ NPL (sig)} + 0.000541 \text{ CAR (unsig)} - 0.002385 \text{ EQRWA(unsig)} - 0.003189 \text{ RWAI (sig)} - 0.000032 \text{ GASSETS (unsig)} \tag{6}$$

Based on the above regression results (Table 6), the following model equation is obtained:

$$\text{ACT\_GR} = -0.060437 \text{ (unsig)} + 0.002464 \text{ COSTINCRATIO (sig)} - 0.036263 \text{ COASSRATIO (sig)} - 0.336504 \text{ REV\_RISK (sig)} + 0.058926 \text{ LDR (sig)} + 0.001188 \text{ NPL (unsig)} + 0.001650 \text{ CAR (unsig)} - 0.001856 \text{ EQRWA (unsig)} + 0.000291 \text{ RWAI (unsig)} - 0.006747 \text{ GASSETS (sig)} \tag{7}$$

Table 6 shows that operational risk effects sustainable growth rate and actual growth, but there is a different effect. Equation (7) shows that operational risk has a negative effect on SGR and a positive effect on actual growth. This means (a) that the higher the operational risk, the lower the determination of long-term sales growth targets, (b) the opposite of actual growth, which indicates the higher the operational risk, the higher the sales growth. When the operational risk is calculated by operating costs against total assets, it shows that (c) the higher the operational risk, the higher the SGR, and the higher the operational risk, the higher the actual growth rate.

As explained earlier, operational risk is a way to measure the level of efficiency, the lower the risk means the more efficient the company. This phenomenon shows that the more efficient means the better the company's performance [75,76] and the company can set a higher operating income target. However, the dependent variable ACT_GR shows that the more inefficient the company is, the higher the operating income, or the additional operational costs are greater than the additional operating income. This is contradictory to the concept of the law of diminishing returns [86]. This is in line with the facts described in Table 3, which shows that there is a significant difference between actual growth and sustainable growth, namely between operational income target policies and actual growth. Two things can be explained in this case: (1) This phenomenon is also an indication of the

inability or failure of internal processes to adapt to external events [74]. When conducting incident research what matters is the COVID-19 pandemic. (2) the occurrence of a conflict of objectives between the desires of shareholders and managers as can be explained by the agency problem [87–89].

The positive relationship between asset efficiency shows a positive effect on both AGR and ACT_GR, meaning that the more inefficient the company is in asset utilization, the higher the company will set the SGR target policy. This is in line with the company's efforts to increase the capacity of asset utilization by increasing the actual growth rate. This finding is inconsistent with [7,16,18,90] The implications of this finding give different meanings when using different efficiency measurement ratios, but both can be explained rationally and logically.

Table 6 and Equation (7) detailing the effect of business risk on SGR and ACT_GR show different findings. There is a positive effect of business risk on SGR and conversely, there is a negative effect of business risk on ACT_GR. This finding shows that the higher the business risk, the higher the policy for setting growth targets, on the contrary, the higher the business risk, the lower the actual growth rate. These findings are in line with [77–79] which show that business risk is a threat from external factors and that the company's inability to overcome the dynamics of competition causes sales to decline. However, the rationality of shareholder interests shows that the higher the business risk, the higher the determination of internal funding policies as seen from the higher SGR. This is also in line with the long-term strategy [16,34,91–93].

The effect of liquidity risk on SGR and ACT_GR as measured by LDR and NPL shows different results. The effect of LDR on SGR and ACT_GR is significantly positive. However, the effect of NPL on SGR is significantly negative and does not affect ACT_GR. This finding indicates that the higher the liquidity risk, the higher the growth of sustainable operating income and the higher the actual operating income of banks. This is possible because LDR in the banking sector has been stipulated by regulation. As explained by [25,80], liquidity risk is the inability of a bank to pay its obligations immediately in the short term. The BASEL III committee banks must maintain a certain level of liquidity to avoid the occurrence of a lack between depositors and bank companies [94–96]. This finding provides the facts that (1) banks consider prudential principles through dividend policies and financial policies which are reflected in the higher SGR and (2) the higher the liquidity, the higher the actual growth rate, indicating the behavior of banks to carry out an aggressive strategy. The rationale for the negative influence of NPL on SGR shows the fact that the higher the NPL as an indication of poor performance, the company will increase caution in its financial policies by setting a higher SGR, in line with [16,54,56].

Financial risk is the risk that arises from the loss of opportunity to get funds from an investment [97]. This study uses three measurement indicators, namely CAR, EQRWA, and RWAI. The Bank's CAR and RWA are determined by international regulations by the BASEL committee. RWA is associated with the minimum amount of capital that must be owned by a bank based on the bank's risk profile from lending activities as a guarantee for financial institutions to avoid the risk of bankruptcy [98]. The higher the RWA, the greater the company's ability to overcome the potential for bankruptcy risk [85,96]. In this study, CAR does not affect SGR or ACT_GR. This is due to the possibility that the CAR is relatively the same for all samples of banks, so the CAR cannot be used as a predictor to determine the growth of operating income. The empirical test results show that only RWAI has a significant negative effect on SGR and has no effect on ACT_GR. This finding means that the higher the financial risk, the lower the SGR. The higher the EQRWA as an indication of the higher the ability of equity in overcoming the risk of bankruptcy. However, this study has not shown any effect of EQRWA on SGR and ACT_GR. This is evidence that the ability of equity to overcome risk has not been explained because the EQRWA for banking samples in several ASEAN countries is relatively the same. The existence of a negative influence between RWAI on SGR can be explained that the higher the RWAI, the greater the bank's ability to overcome bankruptcy risk, this has an impact on the lower retention ratio

target determination (because it has higher cash flow) or the lower SGR [17,67,99]. This is rational and logical to the finding of negative influence between RWAI on SGR.

## 5. Conclusions

This study has concluded several things in the banking sector of several ASEAN countries (excluding Vietnam). Firstly, there are quite significant differences between actual growth, internal growth, and sustainable growth in all the countries studied. This is a reflection on the fact that there is a conflict of objectives between shareholders as SGR targets and what has been achieved by the manager. Secondly, the level of difference is relatively the same in various ASEAN countries. Thirdly, almost all countries experience negative revenue growth rates, but internal growth still shows growth in internal funding sources that can be utilized for asset utilization. Fourthly, SGR policy targets for several countries are relatively the same ranging from 6 to 10 percent and there is no difference in SGR results for using several SGR models, between the Higgins, Ross, and Van-Horne models.

This study reveals that (1) the country with the largest business risk is Thailand, followed by Indonesia, and the lowest business risk is the Philippines and Malaysia. (2) the largest operational risk is Indonesia, followed by the Philippines, and the country with the most efficiency is Singapore followed by Malaysia. (3) the country with the highest liquidity risk from non-performing loans is Thailand, followed by Indonesia, and the country with the lowest liquidity risk is Singapore. (4) the lowest financial risk is Singapore and the highest financial risk is the Philippines, followed by Indonesia. Of the total risks, the country with the greatest risk is Indonesia, followed by Thailand, the Philippines, Malaysia, and Singapore.

The question of how the banking industry maintains sustainability in conditions of facing various risks is answered by the finding that the higher the risk faced, the higher the SGR policy because the potential risk requires companies to be careful in external funding. For operational risk, there are different findings between cost efficiency and asset utilization efficiency, and each effect on SGR and AGR_GR. Cost efficiency has a negative effect on SGR and asset utilization has a positive effect on SGR. On the other hand, asset utilization has a positive effect on SGR and has a positive effect on AGR_GR. It concludes that (1) consideration is needed in using operational risk measurement indicators; (2) there is inconsistency in the determination of dividend policy for internal funding that can respond to dynamic and fluctuating operating income. This finding implies that there is a need for balanced growth that balances the interests of the return on equity of shareholders with the return on company assets through dividend policy and capital structure.

Business risk has a positive effect on SGR and a negative effect on ACT_GR. This finding concludes that there is a precautionary principle from shareholders and managers. When the business risk is high, SGR becomes more stringent for shareholders to set dividend policy and capital structure. The existence of a positive relationship between business risk and ACT_GR concludes two things, namely (1) high risk, high return; and (2) modeling bias relationships because risk measurement uses the standard deviation of actual operating income (ACT_GR).

Liquidity is measured by LDR and NPL having a different effect on SGR and ACT_GR. LDR has a positive effect on SGR and ACT_GR, whereas NPL has a negative effect on SGR and the effect of NPL on ACT_GR has not been explained. These findings conclude (1) that the banking sector has good liquidity performance in line with the SGR policy target and (2) poor NPL performance is followed by the precautionary principle by shareholders by increasing SGR to motivate management.

Financial risk measured by CAR, EQRWA, and RWAI, has a different effect on SGR and ACT_GR. Only RWAI has a significant negative effect on SGR. Empirical facts show that the greater the bank's ability to overcome the risk of bankruptcy, the lower the SGR, meaning the higher the bank's capital adequacy, the higher the dividend payout ratio policy compared to the retention ratio.

This study proves that there is a difference in risk and SGR performance for each country in ASEAN. The relationship among internal growth, actual growth, and sustainability growth proves there is difference for Indonesia, the Philippines, and Singapore but there is no difference between Malaysia and Thailand. Singapore has the lowest total risk compared to the other four countries. Indonesia has the highest total risk. This difference consequently will make ABIF encourage members of ASEAN countries to have common goals to use SGR as a measure of sustainable finance.

The various findings above contribute to the knowledge of financial management in the banking sector in terms of determining dividend policy, and financial and operational policies as well as bridging conflicting objectives between managers and shareholders. Of course, this will have implications for the practice of financial control for shareholders, and how to maintain and set sustainable growth targets in conditions facing various risks in the banking sector. In dealing with uncertainties and various risks that occur, especially in the banking industry that is integrated into the ASEAN Region, it is important to apply the SGR model as a measure of bank performance that pays attention to sustainability, especially the implementation of BASEL III.

The limitation of this study is that it only uses relatively limited data and does not include Vietnam, due to data access problems. Sample selection is only based on data available in the Bank Focus Data Base. For future research, this will take into account the financial characteristics of each country including banking size, regulations at each bank, as well as macro factors that affect banking risk.

**Author Contributions:** Conceptualization, S.; Resources, Y.S.; Writing—review & editing, I.; Supervision, F.J. All authors have read and agreed to the published version of the manuscript.

**Funding:** This research was funded by the DIPA Budget of the Universitas Sriwijaya Public Service Agency for Fiscal Year 2021 No. SP DIPA-023.17.2.677515/2021, 23 November 2021 by the Rector's Decree Number: 0014/UN9/SK.LP2M.PT/202I 25 May 2021.

**Institutional Review Board Statement:** There is no Institutional Review Board.

**Informed Consent Statement:** No research article describing the study.

**Data Availability Statement:** All of authors of this article have not published in MDPI Journal.

**Conflicts of Interest:** The authors declare no conflict of interest.

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
