# Peer review of "Banking Industry Sustainable Growth Rate under Risk: Empirical Study of the Banking Industry in ASEAN Countries"

_sustainability, doi:10.3390/su15010564_

Round 1
Reviewer 1 Report
There is no clear focus discussion of why SGR is important and what are the current studies on SGR IN BANKS, and where are the research gaps you are addressing. The paper needs to be completely rewritten.
The links between SGR and bank performance are not clearly stated.
The definitions of various types of risks mention are inaccurate and needs revisiting, especially when linked to SGR.
Why Banks in ASEAN COUNTRIES? There is obviously going to be differences and how do you control for these differences.
Author Response
Response to Reviewer 1 Comments
Point 1: There is no clear focus discussion of why SGR is important and what are the current studies on SGR IN BANKS, and where are the research gaps you are addressing. The paper needs to be completely rewritten.
Response 1: Please provide your response for Point 1. (in red)
We have corrected and explained why SGR is important and how the current studies on SGR in Banks are, in the abstract & introduction at points 39 to 71, and in the Conclusion points 685 to 691.
Point 2: The links between SGR and bank performance are not clearly stated.
Response 2:
The links between SGR and bank performance have been explained at point 59 sd 71.
Point 3: The definitions of various types of risks mention are inaccurate and needs revisiting, especially when linked to SGR.
Response 3:
The Definition of various types of risks have been corrected in Table: 1
Poin 4: Why Banks in ASEAN COUNTRIES? There is obviously going to be differences and how do you control for these differences?
Response 4: can be seen at point 434-437
This can happen because of the dominance of external funding sources compared to internal funding. Therefore, if this happens, it is necessary to balance growth (balance growth), through restructuring funding sources by increasing retained earnings or efficient use of assets in the banking industry of ASEAN countries.
Reviewer 2 Report
The subject of the article is timely and is interesting for research.
In general, the article has an acceptable structure, but some parts need improvement.
The parentheses can be omitted in the title, and the two parts can be separated by a period.
It is unclear how many authors there are and what their affiliations are.
The abstract is not well structured. The background/broad context of the research and the method used are missing. The results obtained should be synthesized.
The Introduction section is quite general. Besides the general background, it should be included the literature gap, short information on the method and paper structure.
The role of the Literature review section is not to present the formulas used in other research, but to position the research in the specialized literature, to identify the research gap. I recommend the authors to revise this part.
Please explain how the elements mentioned at lines 28-29 are integrated into research.
What are the limitations of the study?
Author Response
Response to Reviewer 2 Comments
Point 1:
In general, the article has an acceptable structure, but some parts need improvement
Response 1:
The improvement has been done using correct spelling.
Point 2: The parentheses can be omitted in the title, and the two parts can be separated by a period.
Response 2: has been improved according to reviewer command.
Point 3: It is unclear how many authors there are and what their affiliations are.
Response 3: has been revised and the author has been added.
Point 4: The abstract is not well structured. The background/broad context of the research and the method used are missing. The results obtained should be synthesized.
Response 4: Abstract has been edited and corrected that make it more specific and structured.
The background has been corrected at point 39-71. Research method used has been explained clearer at point 343-392. The research findings have been synthesized at point 558-631.
Author Response
Response to Reviewer 3 Comments
Point 1:
Response 1:
- I have red the paper and follow several formats in it specifically in area of research design. However, for modelling not all suggestions can be followed due to the difference issues.
- Improved research method has been done.
- Arguments and more specific discussion have been added referring to previous studies at point 557-631.
Reviewer 4 Report
Thank you for submitting your paper for consideration. After carefully reading the paper, This manuscript is “Banking Industry Sustainable Growth Rate Under Risk (Empirical Study of the Banking Industry in ASEAN countries)”. The following issues need to be addressed before publication:
1. Abstract: While the author presents the Abstract, answer the questions carefully: What problem did you study and why is it important? What methods did you use? What were your main results? And what conclusions can you draw from your results? Please make your abstract with more specific and quantitative results while it suits broader audiences. Although some steps have been done following the above suggestion, the revised abstract is still necessary. Besides, please be careful to use the words "scientifically", "systematically," etc.
2. The authors should reorganize the structure of the introduction section to thoroughly express the aspects of this study, including the background, current progress, motivation, research question, objective, contribution, etc.
3. Motivation and contribution need to be strengthened.
4. The literature review should provide a basis for the study by substantiating the research gap. When reviewing the literature, I suggest that you expand your review by citing the most recent study; https://doi.org/10.1007/s11356-022-19763-1
5. Please include new sections to describe this aspect, such as the conclusion, limitations, and suggestions for future research. This section of the manuscript is still weak. Include implications for policymakers as well.
6. Please check the manuscript again for errors.
Author Response
Response to Reviewer 4 Comments
Point 1: 1. Abstract: While the author presents the Abstract, answer the questions carefully: What problem did you study and why is it important? What methods did you use? What were your main results? And what conclusions can you draw from your results? Please make your abstract with more specific and quantitative results while it suits broader audiences. Although some steps have been done following the above suggestion, the revised abstract is still necessary. Besides, please be careful to use the words "scientifically", "systematically," etc.
Response 1:
- Abstract has been clearly explained structurally and sistematically.
- Several corrections and explainations have been added.
- Several improvement has been added at the background at 39-71. Reseach method has been explained at point 343-392. Research findings has been synthesized at point 558-631.
Point 2: . The authors should reorganize the structure of the introduction section to thoroughly express the aspects of this study, including the background, current progress, motivation, research question, objective, contribution, etc.
Response 2: has been corrected at the introduction and the asphect of the study of risk management and growth management are described at the background.
Point 54-58
The challenge for the banking industry is efforts to minimize risk and increase revenue, as a basic concept in the financial literature. The financial literature has explained that high risk, and high return, but risks that are too high can cause corporate bankruptcy or bankruptcy, for example, high financial risk, besides that high credit risk and liquidity risk can disrupt banking integration and stability in ASEAN countries.
Poin 69-71
Determination of SGR is very important for companies because it has two reasons, first as a measurement of company performance and second as a means of controlling shareholders to control their equity.”
Point 220-227
Banking integration in the association of ASEAN countries requires a balanced performance among its members to compete globally. Therefore it is important to explain how the risks and levels of sustainable growth are in the banking industry in ASEAN countries because this can disrupt the soundness of banks and have an impact on economic growth in the ASEAN region. Besides that, this research will prove the impact of risks (business risk, operational risk, liquidity risk, and financial risk) on sustainable growth rates. These mentioned risks are important risks in the banking world as well as a measure of performance and corporate value.
Point 3: Motivation and contribution need to be strengthened.
Response 3:
- motivation of our research
Point 328-332
Risk sensitivity to SGR is very crucial to bridge the relationship between shareholders and managers, in terms of evaluating company performance. Therefore, the focus of this research, apart from describing several SGR measurement models and comparing them with actual growth, also predicts how sensitive various important risks are to SGR in the banking sector of ASEAN.
- Contribution
This study proves that there is a difference in risk and SGR performance for each country in ASEAN. The relationship among internal growth, actual growth, and sustainability growth prove there is the difference for Indonesia, the Philippines, and Singapore but there is no difference between Malaysia and Thailand. Singapore has the lowest total risk compared to another four countries. Indonesia has the highest total risk. This difference consequently will make ABIF encourage members of ASEAN countries to have common goals to use SGR as a measure of sustainable finance
Point 4:. The literature review should provide a basis for the study by substantiating the research gap. When reviewing the literature, I suggest that you expand your review by citing the most recent study; https://doi.org/10.1007/s11356-022-19763-1
Response 4: Yes I do, at the most of the references.
Point 5:. Please include new sections to describe this aspect, such as the conclusion, limitations, and suggestions for future research. This section of the manuscript is still weak. Include implications for policymakers as well.
Response 5
Yes, I do
- The Conclusion: at point 633-690 in the article
- The Implication: at point 692-700
- Limitation and Future Research Agenda: at point 703-709
Poin 6: Please check the manuscript again for errors.
Response 6: Yes I do
Round 2
Reviewer 2 Report
Even if the authors brought some improvements, some revisions are still necessary.
The abstract is too long and is not systematized, synthesized.
Sections 7 and 8 are not needed. The information contained here should be included in the Conclusions section.
Author Response
- Abstract has been shortened and structurally rewritten from 282 word into200 words.
- Section 7 Limitation and Section 8 Implication have been omitted and moved into Section 6 Conclution.

Reviewer 3 Report
Thank you, you response my recommendations successfully.
Author Response
Thank you very much for your kindness